# Novel Non-Metal Cation (NMC) Pentaborate Salts of Some Amino Acids

**DOI:** 10.3390/molecules24152790

**Published:** 2019-07-31

**Authors:** Ümit Sızır, Ömer Yurdakul, Dursun Ali Köse, Fatih Akkurt

**Affiliations:** 1Department of Chemistry, Hitit University, Ulukavak, 19030 Çorum, Turkey; 2Department of Chemical Engineering, Gazi University, Maltepe, 06570 Ankara, Turkey

**Keywords:** non-metal cation (NMC), pentaborate, ^11^B-NMR, thermal analysis, hydrogen storage

## Abstract

Non-metal cation (NMC) pentaborate structures, in which some amino acids (valine, leucine, isoleucine, and threonine) were used as cations, were synthesized. The structural characterization of molecules was carried out by elemental analysis, FT-IR, mass, ^11^B-NMR, and thermal analysis (TGA/DTA) methods. The hydrogen storage capacity of molecules was also calculated by taking experimental results into consideration. The FT-IR spectra support the similarity of structures. The characteristic peaks attributable to pentaborate rings and amino acids were observed. When thermal analysis data were examined, it was observed that pentaborate salts gave similar degradation steps and degradation products. As a final degradation product of all thermal analysis experiments, a glassy form of B_2_O_3_ was observed. The valine pentaborate is the most thermally stable. Also, the amounts of hydrate water outside the coordination sphere of the compounds were determined by thermal analysis curves. The peaks of boric acid, triborate, and pentaborate structures were obtained in ppm with the ^11^B-NMR results of synthesized pentaborate compounds. With powder X-ray spectroscopy, all structures were found to be crystalline but not suitable for single-crystal X-ray analysis. The molecular cavities of the compounds detected by BET were found to be 3.286, 1.873, 2.309, and 1.860 g/cm^3^, respectively. A low number of molecular cavities can be interpreted in several existing hydrogen bonds in structures. The hydrogen storage capacities of the molecules were found to be in the range of 0.04 to 0.07% by mass.

## 1. Introduction

Borates are generally classified as anhydrous mixed metal oxides or hydrated metal borates and the structural properties of about 200 borate minerals are known [1,2]. Some borate structures synthesized in the laboratory form the separated borate or poly-borate anions. However, most of them reveal more condensed anions that form various chains, layers, and nets [3,4,5,6]. In the studies conducted, it was determined that the opposite metal ions in the structures of the boroxol (B_3_O_3_) compound affected the Lewis acidity in various ways. It was also found that boric acid used in borate structures has a four-coordinate position with a strong Lewis acidic character [7,8,9,10].

Due to the rich structural properties of boron-oxygen-containing compounds, there are many applications in the industrial field. The poly-borate-containing compounds are used in conjunction with non-linear optical materials (NLOs) to obtain luminescent materials and lithium electrode batteries [11,12]. Because of these thermal properties of poly-borate compounds, it has become the focus of interest in both industrial and academic studies [13,14,15,16,17,18,19]. One of the most important of these borate compounds obtained as a result of the studies in this laboratory is the anionic pentaborate and hexaborate salts, which contain non-metal compound as a complementary cation. Structural characterizations of some borate materials obtained by the synthesis of these non-metal cation borate salts have been made, and some of these borate compounds are: [B_3_O_3_(OH)_4_]^−^, [B_4_O_5_(OH)_4_]^2−^, [B_5_O_6_(OH)_4_]^−^, [B_7_O_9_(OH)_5_]^2−^, [B_8_O_10_(OH)_6_]^2−^, [B_9_O_12_(OH)_6_]^3−^, [B_14_O_20_(OH)_6_]^4−^, and [B_15_O_20_(OH)_8_]^3−^ [20,21,22].

The chemical interaction potentials of non-metal cation compounds with borate anions are different from metal cations. According to the spherical shape of metal cations, the non-spherical metal cationic compounds differ in terms of accepting electrons from the oxygen atom and forming H-bonds [23,24]. This electronic and steric factor significantly affects the structure of non-metal cation borate compounds. For example, ammonium-borite mineral-free borate minerals, [NH_4_]_3_ [B_15_O_20_(OH)_8_].4H_2_O and larderit [NH_4_][B_5_O_7_(OH)_2_]H_2_O poly-borate minerals having anionic borate structures coupled with ammonium cation, and poly-borate systems from amine group-containing compounds, such as guanidinium and imidazolium [B_9_O_12_(OH)_6_]^3−^, are single metal non-cationic nonaborate compounds containing isolated nonaborate anion [2,24,25].

The metal or non-metal pentaborate structures are highly stable and transformed into a cyclic structure by the polymerization reaction resulting from dehydration in the basic medium of aqueous solutions (containing -OH) [26,27,28,29,30]. The non-metal hexaborates are also formed by the reaction of excess boric acid. Pentaborate include neutral trigonal BO_3_ and interconnected anionic tetragonal BO_4_^−^. Hexaborates contain trigonal BO_3_ and interconnected tetragonal BO_4_^−^ structures as well as excessive amounts of trigonal B(OH)_3_. When the metal cation-stabilized pentaborate and hexaborate anions contain non-metal cation compounds as the stabilizing ion, more porous and channel-free compounds are obtained. In the synthesis of non-metal cation pentaborate and hexaborate salts, aqueous or non-aqueous boric acid may be used [31].

Commonly, the organic compounds used in the synthesis of non-metal cation pentaborate salts include amine groups. The reason for this is that the commercial delivery of compounds containing 1°, 2°, and 3° amine groups is easy and cheap. However, one disadvantage is that the free amine groups are very basic in aqueous solutions dominated by hydroxide ions (where the free protonated free base in amino acids is pKa greater than 10). In the synthesis of borate salts which do not contain metal cation, it is necessary to use benzene (or toluene) as a solvent to ensure that the environment is weakly basic and to remove water with azeotropic distillation, if there is water in the environment [26,27,28,29,32].

When quaternization of the free amine groups is generally carried out using Ag_2_O or ion exchange resins, a cationic 4° ammonium salt can be obtained [33]. The carbonate compounds of some organic cations can be found commercially or synthetically [34].

In anhydrous solutions, the slow hydrolysis of boric acid in the presence of base B in the form of B(OMe)_3_ is another alternative method [35]. It has been found that this method is very successful, and the original pentaborate ester anions’ [C_8_H_6_(NMe_2_)_2_H] [B_5_O_6_(OMe)_4_] crystals can be obtained [36].

Initially, the structures of pure tetraalkylammonium materials obtained with organic bases interacting with anions via H-bonds were observed. When we investigated the interaction of the steric -CH groups, the -OH groups were H-link donors or receptors, while the -NH sites were generally H-link donors. The non-metal cation containing pentaborate structures as a counter-ion were obtained using the quaternary ammonium compound. However, very few structural characterizations have been examined. Recently, some metal cation/anion pentaborate salts have been characterized [2,24,26,29,30,31,32,34,35,37].

Ammonium pentaborate tetrahydrate (NH_4_)_2_O·5B_2_O_3_·8H_2_O (NH_4_B_5_O_8_·4H_2_O), one of the pentaborate salts, is one of the borate compounds that are commercially available in many industries (electrolytic capacitors, corrosion inhibition, fire retardant, welding, soldering, and brazing fluxes) [38].

In this study, the non-metal cation pentaborate structures of some amino acids (valine, leucine, isoleucine, and threonine) were synthesized. Zwitterion amino acids were used as counter ions of anionic pentaborate structures. The newly synthesized molecules were structurally characterized by elemental analysis, FT-IR, mass, ^11^B-NMR, and thermal analysis (TGA/DTA) methods. Molecular spaces of the structures were analyzed by BET analysis. The molecular spaces of the structures were analyzed by BET analysis. The hydrogen storage capacities of the molecules were also calculated considering the experimental results.

## 2. Results and Discussion

### 2.1. Elemental Analysis

The elemental analysis data of pentaborate salts containing organic compound as the counter cation are given in Table 1. According to the elemental analysis data, the experimental values are in agreement with the theoretical values. Some deviations in the values are thought to be caused by organic and inorganic impurities that cling onto the structure in the synthesis of pentaborate salts. According to these values, we can say that the highest yield was found for isoleucine pentaborate (**III**) at 79%, while the lowest yield was found for valine pentaborate (**I**) salt at 65%. The melting points (MPs) of pentaborate salts containing the organic compound as a complementary cation were determined, and they are given in Table 1. Valine pentaborate had the highest melting point of 134 °C, while other pentaborate structures showed similar melting temperatures (121–123 °C). All compounds were white.

### 2.2. FT-IR Spectroscopy

The FT-IR spectra of valine, leucine, isoleucine, and threonine pentaborate salts are shown in Figure 1, and significant peaks of the spectra are given in Table 2. The strong asymmetric peaks attributable to the B-O bond of BO_3_ in the structure of the synthesized valine, leucine, isoleucine, and threonine pentaborate salts were observed in the 1439, 1444, 1432, and 1420 cm^−1^ regions, respectively. The symmetrical peaks of the B-O bond of the BO_3_ structure of leucine, isoleucine, and threonine pentaborates appeared in 1329, 1334, and 1333 cm^−1^. The symmetrical peak of the valine pentaborate was not determined by the strong asymmetric peak [24,39,40,41,42].

The B-O bond in the BO_4_^−^ the structure of non-metal cation pentaborate salts observed strong asymmetric peaks in the 1026, 1035, 1024, and 1032 cm^−1^ regions for valine, leucine, isoleucine, and threonine pentaborates, respectively, whereas 927, 923, 929, and 927 cm^−1^ symmetrical peaks were also determined. In-plane bending asymmetrical/symmetrical peaks of the B-O-H bond were found for valine, leucine, isoleucine, and threonine pentaborates, respectively: 1199_asym_/1166_sym_, 1194_asym_/1167_sym_, 1193_asym_/1163_sym_, and 1195_asym_/1167_sym_ cm^−1^ were determined.

Out-of-plane bending peaks of B–O bonding of the BO_3_ structure were found to be 698 cm^−1^ for valine pentaborate salt, 682 cm^−1^ for leucine pentaborate, 696 cm^−1^ for isoleucine pentaborate, and 696 cm^−1^ for threonine pentaborate. The bending peaks, which are attributable to the NH_2_ groups of the amino acids, which are cationic present as stabilizing ions in the structures, of the 1666 cm^−1^ (**I**), 1680 cm^−1^ (**III**), and 1678 cm^−1^ (**IV**) regions emerged, respectively. The N-H vibrations were observed at the 3214, 3216, and 3213 cm^−1^ wavenumbers for the stretching peaks. The N-H peak of the leucine pentaborate salt could not be detected due to interference with the strong and broad -OH^−^ peak [24,43].

The -OH groups of aqua molecules and B–OH structures in the aminoacid pentaborate salts were found to have strong and broad stretching vibrations at 3378 cm^−1^ (**I**), 3439 cm^−1^ (**II**), 3383 cm^−1^ (**III**), and 3446 cm^−1^ (**IV**).

### 2.3. ^11^B-NMR Spectroscopy

In Appendix A, the ^11^B-NMR peak values are given in ppm, and the ^11^B-NMR spectra of pentaborate structures are shown in Figure 2. The shift values of the obtained ^11^B are consistent with the results of similar studies in the literature [44,45,46]. According to the results, the highest numerical value of the boric acid peak was observed in valine pentaborate salt with 19.37 ppm, and the lowest quantitative value was observed in leucine pentaborate salt with 17.25 ppm. It can be said that the boric acid peaks are caused by the excessive amount of boric acid used in the synthesis of non-metal cation pentaborate. The chemical shift value for the triborate structure was observed in the pentaborate salt of threonin, with a maximum of 13.32 ppm, while the smallest chemical shift value was observed in valine pentaborate salt, with 12.53 ppm. When the chemical shift values of the pentaborate structure are compared, it can be said that the highest valine pentaborate salt is 1.17 ppm, and the lowest amount of threonine pentaborate salt is 1.00 ppm. The salts dissolve in aqueous solution and the ^11^B-NMR is consistent with the decomposition of salts as facile equilibria, involving fast exchanging three-coordinate and four-coordinate boron centers, is rapidly established. The triborate(1-) and monoborate(1-) species observed were generally in solutions arising from pentaborate(1-) anions [44,47].

### 2.4. Thermal Analysis

*Valine pentaborate salt:* The thermal analysis curves of the valine pentaborate salt (TG-DTG and DTA) are shown in Appendix A (Appendix A), and the details of the degradation steps of the decomposition products are given in Table 3 In the first step of the valine pentaborate salt decomposition, 1-mole hydrate water outside the coordination sphere is evaporated from the structure (Equation (1)):
(1)$$C_{5}H_{17}B_{5}{\mathrm{NO}}_{13(s)}\overset{35–112\;{}^{{}^{\circ}}C}{\to }C_{5}H_{15}B_{5}{\mathrm{NO}}_{12(s)}+H_{2}O_{\left(g\right)}^{↑}$$


The -OH groups in the ring pentaborate structure in the valine pentaborate salt in the 113 to 156 °C temperature range are separated in 2 mol. At the same time, the amine group in the structure of the valine amino acid compound appears to separate in the form of 1 mol of ammonia (Equation (2)):
(2)$$C_{5}H_{15}B_{5}{\mathrm{NO}}_{12(s)}\overset{113–156\;{}^{{}^{\circ}}C}{\to }C_{5}H_{8}B_{5}O_{10(s)}+2H_{2}O_{\left(g\right)}^{↑}+{\mathrm{NH}}_{3(g)}^{↑}$$


In the temperature range of 157 to 580 °C, 1 mol of the organic part of the remaining structure is degraded, and the black glassy 5/2 molar B_2_O_3_ compound remains as the degradation product. The black color of the residual product can be attributed to the carbonized residue, which cannot burn (Equation (3)):
(3)$$C_{5}H_{8}B_{5}O_{10(s)}\overset{157–580\;{}^{{}^{\circ}}C}{\to }{\mathrm{CO}}_{2}/CO_{\left(g\right)}^{↑}+H_{2}O_{\left(g\right)}^{↑}+5/2\;{B_{2}O_{3(s)}}_{↓}$$


The thermal decomposition information of other amino acid pentaborates (leucine, isoleucine, and threonine) is very similar to that of valine pentaborate salt. Boron oxide remains the final degradation product in all of them, while degradation begins with dehydration. The thermal decomposition steps and decomposition products for all compounds are detailed in Table 3. Thermal decomposition curves are given in the Appendix A.

### 2.5. Powder X-ray Diffraction Analysis

In Figure 3, the powder X-ray diffraction patterns of pentaborate salts are shown. The information about the structural characterization of the powder by X-ray diffraction analysis was obtained. All of the molecules were found to be in crystalline form. The structural formulas could not be proposed precisely because single crystal forms could not be isolated. The crystallinity of the synthesized molecules is high, but the threonine pentaborate has poorer crystallinity compared to other structures when the structures are compared. The peaks marked in the spectra indicate the formation of pentaborate rings [48]. Based on the literature information, we can say that when the powder X-ray spectra were examined, it supports the formation of pentaborate pentane in the structure indicated by the ^11^B-NMR spectrum.

### 2.6. Mass Spectrometry Analysis

The mass analysis of aminoacid pentaborate salts was made by GC-MS. As a result of the analysis, a graph of the relative efficiency against the mass/charge (*m*/*z*) ratio for each pentaborate salt was obtained, as shown in Appendix A.

In the valine pentaborate graph, boron hydroxide at 44.02 *m*/*z*, boron tetraoxide at 73.09 *m*/*z*, boron tetrahydroxide at 76.01 *m*/*z*, valine amino acid compound at 116.06 *m*/*z*, and 1/2 mol decaborate at 174.17 *m*/*z* can be interpreted as ion peaks. The peaks attribut2able to the molecular ion peaks of the valine pentaborate and pentaborate compounds were determined at a ratio of 344.01 *m*/*z* and 213.05 *m/z*, respectively. In the leucine pentaborate graph, boron hydroxide at 44.17 *m*/*z*, boron tetraoxide at 73.00 *m*/*z*, boron tetrahydroxide at 76.25 *m*/*z*, valine amino acid compound at 130.32 *m*/*z*, and 1/2 mol decaborate at 174.14 *m/z* can be interpreted as ion peaks. The peaks attributable to the molecular ion peaks of the leucine pentaborate and pentaborate compounds were determined at a ratio of 367.02 *m/z* and 207.02 *m*/*z*, respectively. In the isoleucine pentaborate graph, boron hydroxide at 44.01 *m/z*, boron tetraoxide at 73.15 *m*/*z*, boron tetrahydroxide at 76.04 *m*/*z*, valine amino acid compound at 130.07 *m*/*z*, and 1/2 mol decaborate at 174.14 *m*/*z* can be interpreted as ion peaks. The peaks attributable to the molecular ion peaks of the leucine pentaborate and pentaborate compounds were determined at a ratio of 367.11 *m*/*z* and 213.15 *m*/*z*, respectively. In the threonine pentaborate graph, boron hydroxide at 44.04 *m*/*z*, boron tetraoxide at 73.05 *m*/*z*, boron tetrahydroxide at 76.99 *m*/*z*, valine amino acid compound at 118.06 *m*/*z*, and 1/2 mol decaborate at 174.20 *m*/*z* can be interpreted as ion peaks. The peaks attributable to the molecular ion peaks of the leucine pentaborate and pentaborate compounds were determined at a ratio of 346.21 *m*/*z* and 213.06 *m*/*z*, respectively. The mass load ratio of the leucine pentaborate and isoleucine pentaborate structures is approximately the same due to their isomerism. However, the isolation of isoleucine pentaborate as high as 0.9 *m*/*z* relative to leucine may be attributed to the fact that the -CH_3_ group in the isoleucine is close to the alpha carbon and thus, the ionization is more difficult.

### 2.7. BET Analysis

The surface pore size (BCH) scatter graph and surface area (DFT) scatter graphs were obtained by BET analysis. The surface area values of the pentaborate salts are given in Table 4. According to these data, the highest surface area was observed in the valine pentaborate, while the lowest surface area was found for threonine pentaborate.

### 2.8. Determination of Hydrogen Storage Capacity

The hydrogen storage performance of pentaborate compounds obtained as a result of synthesis was measured experimentally according to high pressure volumetric and mass analysis (HPVA) techniques. A certain amount of samples were taken from these compounds and, in the temperature range of 75 to 100 °C, the hydrogen storage performance at different pressures up to 1 bar and at 77 K was measured after degassing, and activation procedures were performed for 4 to 8 h. The pressure graph of the mass of hydrogenated pentaborate salts showing hydrogen storage capacities is shown in Figure 4.

In Table 5, the hydrogen storage capacities for each molecule sample as the mass and volume are given.

It is clear that the **IV**-labeled threonine pentaborate compound has a higher hydrogen storage capacity than the other samples. Threonine pentaborate can store 0.063% by mass of hydrogen at 77 K and 1 bar pressure. The valine pentaborate compound (**I**) has the lowest capacity, with a hydrogen storage capacity of 0.039% by mass under the same conditions.

## 3. Material and Methods

### Synthesis of Aminoacid-Pentaborates Structures

In total, 0.008 moles of l-valine (1 g) (Sigma-Aldrich) was dissolved in a 50 mL/50% distilled water/methanol mixture (Sigma-Aldrich, İstanbul, Turkey) in a 100 mL beaker. No pH adjustment was made. Then, 0.04 mol of H_3_BO_3_ (Sigma-Aldrich) was added to the solution as a solid. The solution was stirred at room temperature for about 1 h with a magnetic stirrer. Then, a part of the solvent was removed by an evaporator until a hard consistency of the solution was obtained, and was to crystallize at room temperature. The product obtained was dried at 50 °C in a vacuum oven.

The reason for not adjusting the acid-base is that amino acids are present in aqueous solutions (at pH close to neutral pH) in the form of a double ion (zwitterion form) [49]. That is, the amine (–NH_2_) group attached to the α-carbon atom takes the acidic hydrogen (-COOH) of the carboxylic acid, and the carboxylic acid group is present in the anionic state (-COO^−^) because it also gives hydrogen. The amine group that received the hydrogen ion is converted to the cationic state (-NH_3_^+^) (Figure 5).

Two moles of water will react against 1 mol of boric acid, that is, 0.02 mol of water will be reacted to 0.01 mol of boric acid. The boric acid dissolved in water will exhibit the Lewis acid state and will take the -OH^−^ group in water and turn into B(OH)_4_^−^, which has an anionic tetrahedral coordination structure (Figure 6). The limiting component is boric acid.

Based on this information, the amount of boric acid remaining insoluble in the environment is 0.03 moles, and the remaining amount of water is 0.6522 moles. The remaining 0.03 moles of boric acid will decrease the dissolution rate due to the azeotropic mixture. In this way, the Lewis acid character of the −H^+^ cation resulting from the dissolution of B(OH)_3_ leads to the formation of the triple coordinate structure of BO_3_. Therefore, the -COO^−^ a group of the double ionic amino acid will take the -H^+^ cation given to the medium and the amino acid compound will be cationic instead of the double ion. The limiting component is the amino acid, with 0.008 moles. The remaining 0.022 moles of water may remain as hydrate water due to the formation of the pentaborate structure. The counterion [B(OH)_4_]^−^ of the non-metal cation compound (aminoacid) and the partially dissolved B(OH)_3_ will be obtained in the solution medium. As a result of crystallization, we can say that, via water elimination, BO_4_^−^ and BO_3_ bonded, and they formed the pentaborate ring structure (Figure 7).

In these studies, cationic forms of non-metal valine, leucine, isoleucine, and threonine amino acids were used as the counterion of anionic pentaborate rings for the construction of non-metal cation pentaborate salts (Figure 8). The structures and thermal properties of the synthesized pentaborates were investigated using FT-IR (Perkin Elmer Spectrum One B, MA, USA), ^11^B-NMR (Bruker 500 MHz, D_2_O, Germany), Brunauer–Emmett–Teller (BET) surface area analysis (Quantachrome, Autosorb- iQ-MP^®^/XR, Austria), X-ray diffraction (Rigaku Ultima-IV, Japan), melting point (Stuart SMP30 Melt Point, UK) determination, elemental analysis (LECO, CHNS-932, MI, USA), and TGA/DTA/DTG analysis (Shimadzu DTG60H, Japan). At the same time, the hydrogen storage properties of the molecules were determined using a Micromeritics HPVA 100 (High-Pressure Volumetric Analyzer, GA, USA).

## 4. Conclusions

In the synthesis of pentaborate compounds that do not contain the metal cation, the amino acids of valine, leucine, isoleucine, and threonine were used as cations. The structure analysis of each NMC pentaborate compound was carried out. Due to the small volume of organic cations, the hexaborate synthesis was not successful. When the FT-IR spectra were examined, it was determined that the structures showed similarities and the bonding and coordination characters were supported by the infrared spectrum results. When thermal analysis data were examined, it was observed that pentaborate salts gave similar degradation steps and degradation products.

As a final degradation product of all the thermal analysis results, the glassy form of the related B_2_O_3_ was observed, and it was found that the oxygen in the residue product was supplied from boric acid because the process was carried out in a nitrogen atmosphere. The experimental and theoretical results of the thermal analysis showed different characteristics depending on the determined DTA temperatures. The most stable compound was valine pentaborate, and the most unstable compound was leucine pentaborate. At the same time, from the results of the thermal analysis, we can say that the hydrate water outside the coordination sphere of the compounds had different proportions. The peaks of boric acid, triborate, and pentaborate structures were obtained in ppm with the ^11^B-NMR results.

With the use of powder X-ray diffraction measurements, all structures were found to be crystalline but not suitable for single crystal X-ray analysis.

When compared with pentaborate salts synthesized according to their hydrogen storage capacities, the sequence changes were threonine pentaborate > isoleucine pentaborate > leucine pentaborate > valine pentaborate. This is because the threonine pentaborate salt has a lot of pores where hydrogen gas can be captured. There was a difference in the molecular cavity areas of the synthesized compounds and the hydrogen storage results. This is because the BET analysis of the hydrogen storage phase was based on the inert and molar mass of nitrogen with a large nitrogen gas. In this way, hydrogen gas is better adsorbed than nitrogen gas. The hydrogen storage capacities were in the range of 0.04 to 0.07% by mass. According to the literature (MOF-74 Ni or Mg-centered, 1.80%), the hydrogen storage capacities were found to be low [50,51].

However, the composites with metals providing a catalytic effect for hydrogen, such as Pt, Pd may be considered as a way to increase hydrogen storage performance. An example of this is the addition of a Pt-metal to MOF-5, resulting in an approximately 50% increase in hydrogen storage performance at 77 K and a pressure of 1 bar [52].

## Figures and Tables

**Figure 1 molecules-24-02790-f001:**
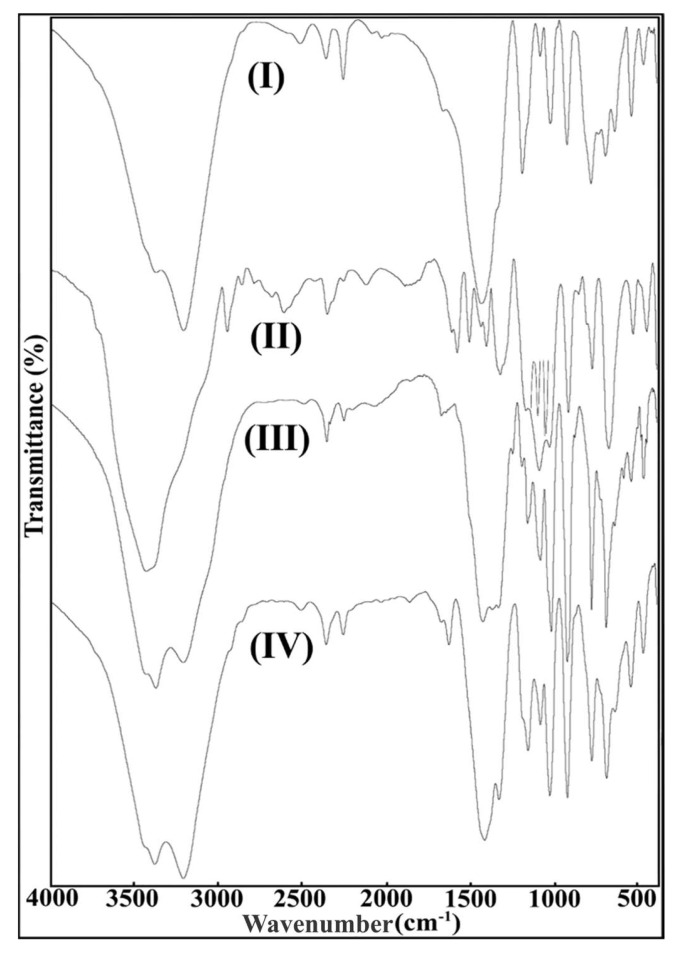
FT-IR spectra of non-metal cation pentaborate structure. (**I**) Valine pentaborate, (**II**) leucine pentaborate, (**III**) isoleucine pentaborate, (**IV**) threonine pentaborate.

**Figure 2 molecules-24-02790-f002:**
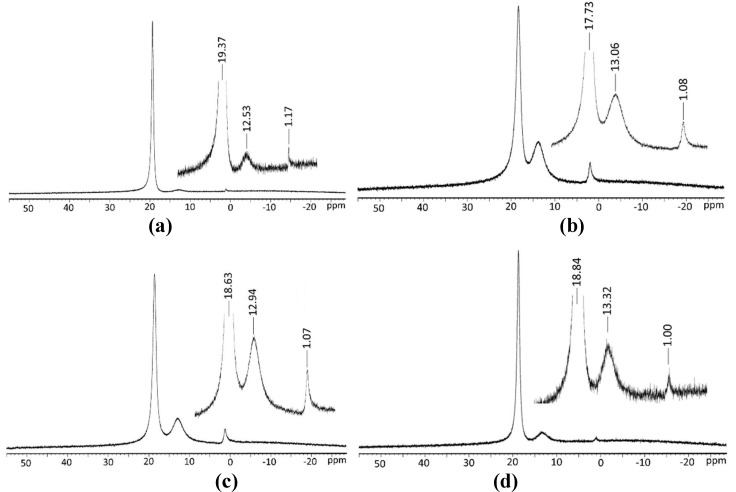
^11^B-NMR spectra of non-metal cations pentaborate salts, (**a**) valine pentaborate, (**b**) leucine pentaborate, (**c**) isoleucine pentaborate, (**d**) threonine pentaborate.

**Figure 3 molecules-24-02790-f003:**
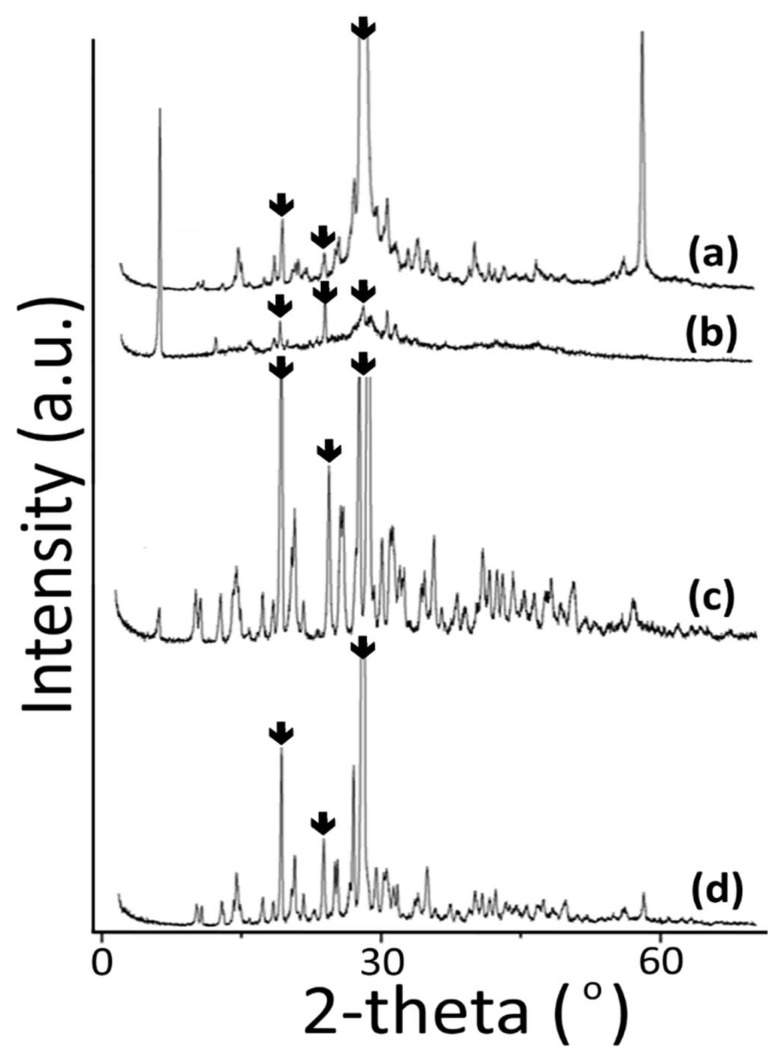
X-ray diffraction spectra of aminoacid pentaborate salts. (**a**) Valine pentaborate, (**b**) leucine pentaborate, (**c**) isoleucine pentaborate, (**d**) threonine pentaborate.

**Figure 4 molecules-24-02790-f004:**
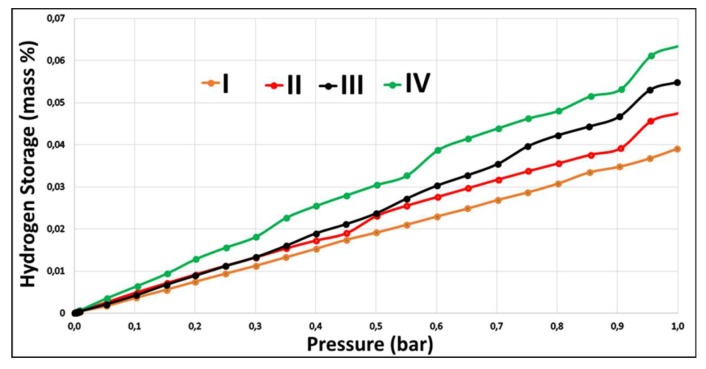
Hydrogen storage capacity graph of the pentaborate compounds. (**I**) Valine pentaborate, (**II**) leucine pentaborate, (**III**) isoleucine pentaborate, (**IV**) threonine pentaborate.

**Figure 5 molecules-24-02790-f005:**
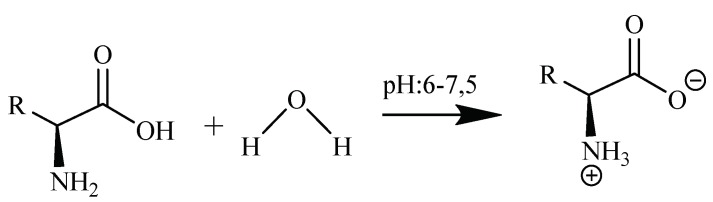
Behavior of amino acids in aqueous solutions.

**Figure 6 molecules-24-02790-f006:**
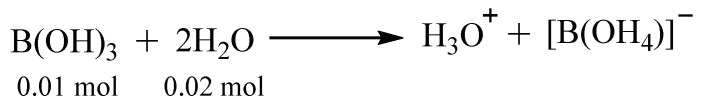
The reaction of boric acid and the water molecule.

**Figure 7 molecules-24-02790-f007:**
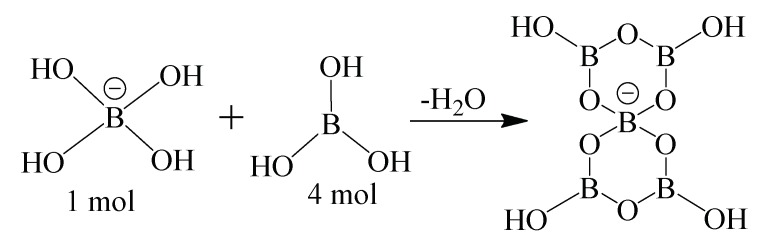
Formation of pentaborate rings.

**Figure 8 molecules-24-02790-f008:**
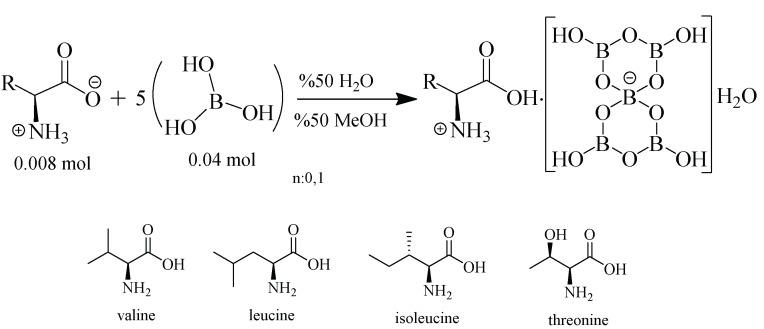
Non-metal cation (NMC) pentaborate synthesis reaction with amino acid residues.

**Table 1 molecules-24-02790-t001:** Analysis data of pentaborate salts containing organic compound as a counter cation.

Compound	Color	MP (°C)	MW g/mol	Yield (%)	Chemical Composition Exp. (Calc.)		
					C	H	N
[C_5_H_11_NO_2_][B_5_O_6_(OH)_4_]H_2_O C_5_H_17_B_5_NO_13_ **(I)**	White	134	345.24	65	18.97 (17.39)	5.41 (4.96)	3.89 (4.06)
[C_6_H_13_NO_2_][B_5_O_6_(OH)_4_]1/2H_2_O C_6_H_18_B_5_NO_12,5_ **(II)**	White	122	368.28	71	19.91 (19.57)	5.62 (5.47)	3.87 (3.80)
[C_6_H_13_NO_2_][B_5_O_6_(OH)_4_]H_2_O C_6_H_19_B_5_NO_13_ **(III)**	White	121	368.28	79	19.69 (19.57)	5.81 (5.47)	3.92 (3.80)
[C_4_H_9_NO_3_][B_5_O_6_(OH)_4_]H_2_O C_4_H_15_B_5_NO_14_ **(IV)**	White	123	356.22	69	13.01 (13.49)	4.88 (4.53)	3.77 (3.93)

**Table 2 molecules-24-02790-t002:** Some important IR peaks of non-metal cation pentaborate salts.

Groups	Valine Pentaborate (I)	Leucine Pentaborate (II)	Isoleucine Pentaborate (III)	Threonine Pentaborate (IV)
ν(-OH)_H2O_	3378	3439	3383	3346
ν(-NH)	3214	-	3216	3213
ν(-NH_2_)	1666	-	1680	1678
ν(-B-O)_BO3_	698	682	696	694
ν(-B-O)_BO3(asym)_	1439	1444	1432	1420
ν(-B-O)_BO3(sym)_	-	1329	1338	1334
ν(-B-O-H)	1199/1166	1194/1167	1193/1163	1195/1167
ν(-B-O)_BO4(asym)_	1026	1035	1024	1032
ν(-B-O)_BO4(sym)_	927	923	929	927

**Table 3 molecules-24-02790-t003:** Some important thermal analysis data of aminoacid pentaborate salts.

Complex	Temp. Range (°C)	DTAmax. (°C)	Removed Group	Mass Loss (%)		Remaining Product (%)		Decomp. Product	Color
				Found	Calc.	Found	Clac.		
C_5_H_17_B_5_NO_13_	35–112	82	-H_2_O	2.03	2.61				White
345.24 g/mol	113–156	144	-2H_2_O, NH_3_	16.72	15.35				
**(I)**	157–580	172, 295, 460	-C_5_H_8_O_2,5_	28.92	29.58	48.97	50.69	5/2 B_2_O_3_	Black
C_6_H_18_B_5_NO_12,5_	37–97	64	-H2O	3.96	4.89				White
368.28 g/mol	98–219	117, 144	-2H_2_O, NH_3_	14.88	14.39				
**(II)**	220–720	256, 582	-C_6_H_9_O_2_	30.17	31.54	45.97	47.52	5/2 B_2_O_3_	Black
C_6_H_19_B_5_NO_13_	31–144	77	-H_2_O	4.02	4.89				White
368.28 g/mol	102–145	130	-2H_2_O, NH_3_	15.92	14.39				
**(III)**	146–712	181, 235, 317, 601	-C_6_H_10_O_2,5_	30.82	31.54	46.19	47.52	5/2 B_2_O_3_	Black
C_4_H_15_B_5_NO_14_	33–90	75	-H_2_O	3.27	4.08				White
356.22 g/mol	91–150	131	-2H_2_O, NH_3_	16.11	15.86				
**(IV)**	152–655	170, 325, 510	-C_4_H_6_O_3,5_	39.07	39.93	38.69	39.66	5/2 B_2_O_3_	Black

**Table 4 molecules-24-02790-t004:** BET analysis surface area results table of pentaborate salts.

Molecules	Surface Area (m^2^/g)
[C_5_H_11_NO_2_][B_5_O_6_(OH)_4_]H_2_O C_5_H_17_B_5_NO_13_ **(I)**	3.286
[C_6_H_13_NO_2_][B_5_O_6_(OH)_4_]1/2H_2_O C_6_H_18_B_5_NO_12.5_ **(II)**	1.873
[C_6_H_13_NO_2_][B_5_O_6_(OH)_4_]H_2_O C_6_H_29_B_5_NO_13_ **(III)**	2.309
[C_4_H_9_NO_3_][B_5_O_6_(OH)_4_]H_2_O C_4_H_15_B_5_NO_14_ **(IV)**	1.860

**Table 5 molecules-24-02790-t005:** Hydrogen storage capacities of pentaborate salts at 77 K and 1 bar.

Molecules	Mass Storage (%)	Volumetric Storage (mL/g)
[C_5_H_11_NO_2_][B_5_O_6_(OH)_4_]H_2_O C_5_H_17_B_5_NO_13_ **(I)**	0.039	0.091
[C_6_H_13_NO_2_][B_5_O_6_(OH)_4_]1/2H_2_O C_6_H_18_B_5_NO_12.5_ **(II)**	0.047	0.115
[C_6_H_13_NO_2_][B_5_O_6_(OH)_4_]H_2_O C_6_H_29_B_5_NO_13_ **(III)**	0.055	0.133
[C_4_H_9_NO_3_][B_5_O_6_(OH)_4_]H_2_O C_4_H_15_B_5_NO_14_ **(IV)**	0.063	0.155

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
