# Peer review of "Novel Non-Metal Cation (NMC) Pentaborate Salts of Some Amino Acids"

_molecules, 2019, doi:10.3390/molecules24152790_

Round 1

Reviewer 1 Report

The science is good, but the usage of English is very poor! For example, the title should read as "Synthetic, Structural and Thermal Characterization of Some Novel Non-Metallocationic (NMCs)
Pentaborates Salts"!  Similarly, there are many more such phrases throughout the text and those must be corrected before publication. 

In addition, I see many relevant literature citations are missing.  I am surprised that the authors are not at all aware of relevant chapters in Boron Science/Chemistry and COMC-III books and journals published recently! All of these missing citations must be included before publication of this work is permitted.

Author Response

The science is good, but the usage of English is very poor! For example, the title should read as "Synthetic, Structural and Thermal Characterization of Some Novel Non-Metallocationic (NMCs)
Pentaborates Salts"!  Similarly, there are many more such phrases throughout the text and those must be corrected before publication. 

Response: - The title of the article was simplified with the proposal of the referee and the English version of the article was reviewed.

In addition, I see many relevant literature citations are missing.  I am surprised that the authors are not at all aware of relevant chapters in Boron Science/Chemistry and COMC-III books and journals published recently! All of these missing citations must be included before publication of this work is permitted.

Response:- The part of the book proposed by the referee related to the pentaborates is included in the references.

Reviewer 2 Report

This paper reports on the synthesis and characterization of pentaborate salts with protonated amino acids. The compounds have been characterized by using several techniques, mainly in the solid state (elemental analysis, IR, Powder X-ray diffraction,  TGA analysis).  The ability of compounds in hydrogen storage has been also evaluated.

The chemistry of pentaborates can be of some interest in the field of inorganic chemistry. On the other hand the paper suffers a number of problems in style and content, making it not suitable for publication in the present form.

The first concern is the really awkward and, sometimes, scarcely comprehensible, English used.

A few examples (but there are many others):

Abstract: “Structural characterization of molecules by elemental analysis, FT-IR, Mass, 11B-NMR and thermal analysis (TGA/DTA) methods”. A verb has been missed here.

“The peaks of boric acid, triborate and pentaborate structures were obtained in ppm with the 11B-NMR results of these pentaborate compounds”   This sentence should be rephrased

Page 3, line 113: “B(OH)4- which has an anionic quadrature coordination structure. What do the authors mean with ‘quadrature’?

The title of chapter 3.2 and the first line of Table 3.5 are not written in English.

A second, and most important, point is purity of compounds.  The 11B NMR spectra show, together with the signal of pentaborate, the signals of triborate and boric acid (Figure 6 and Table 3). The latter two compounds appear to be present in consistent percentage (the authors do not try to quantify the content by signal integration). I do not understand the sense of a detailed characterization study on not pure compounds. The pentaborate salts should be purified before characterization and further studies on gas storages.

Other points.

- The paper is too long and several characterization data could be supplied as supporting information (such as the elemental analysis data and the NMR chemical shift table). The Introduction chapter should be shortened, removing ‘trivial’ discussion points, such as the acid-base behaviour of amino acids.

- Figure 9 (mass spectra) is too small and, therefore, not readable.

- There is no comparison with previously reported pentaborate salts cited in the introduction. It should be of interest to know the differences between the present compounds and pentaborate salts having metal cations as counter ions.

- There almost no experimental data. For instance, which solvent used the authors to collect the 11B NMR spectra?  Did the authors collect NMR spectra on solid compounds?

In summary, I cannot suggest publication of this paper. The compounds should be carefully isolated and purified. The paper should be deeply revised, removing some not useful details and adding some discussion/comparison with similar compounds. Finally, the language used should be deeply revised and the paper should be corrected by a native English speaker.

Author Response

This paper reports on the synthesis and characterization of pentaborate salts with protonated amino acids. The compounds have been characterized by using several techniques, mainly in the solid-state (elemental analysis, IR, Powder X-ray diffraction,  TGA analysis).  The ability of compounds in hydrogen storage has been also evaluated.

The chemistry of pentaborates can be of some interest in the field of inorganic chemistry. On the other hand, the paper suffers a number of problems in style and content, making it not suitable for publication in the present form.

The first concern is the really awkward and, sometimes, scarcely comprehensible, English used.

-        The English language of the article was reviewed as a grammar.

A few examples (but there are many others):

Abstract: “Structural characterization of molecules by elemental analysis, FT-IR, Mass, 11B-NMR and thermal analysis (TGA/DTA) methods”. A verb has been missed here.

-        This sentence has been corrected. (The structural characterization of molecules was carried out by elemental analysis, FT-IR, Mass, 11B-NMR, and thermal analysis (TGA/DTA) methods.)

“The peaks of boric acid, triborate, and pentaborate structures were obtained in ppm with the 11B-NMR results of these pentaborate compounds”   This sentence should be rephrased

-        This sentence has been corrected. (The peaks of boric acid, triborate, and pentaborate structures were obtained in ppm with the 11B-NMR results of synthesized pentaborate compounds.)

Page 3, line 113: “B(OH)4- which has an anionic quadrature coordination structure. What do the authors mean with ‘quadrature’?

-        The word "quadrature" was replaced by the word "tetrahedral" to make it more understandable.

The title of chapter 3.2 and the first line of Table 3.5 are not written in English.

-        Spelling mistakes made by forgetting were corrected.

A second, and most important, the point is purity of compounds.  The 11B NMR spectra show, together with the signal of pentaborate, the signals of triborate and boric acid (Figure 6 and Table 3). The latter two compounds appear to be present in a consistent percentage (the authors do not try to quantify the content by signal integration). I do not understand the sense of a detailed characterization study on not pure compounds. The pentaborate salts should be purified before characterization and further studies on gas storages.

-        An explanation from literature on the referee's criticism was added to the NMR interpretation and the necessary literature resources were added to the references. (The salts dissolve in aqueous solution and 11B-NMR is consistent with the decomposition of the salts as facile equilibria, involving fast exchanging 3-coordinate and 4-coordinate boron centers, is rapidly established. The triborate(1-) and monoborate(1-) species observed are generally observed in solutions arising from pentaborate(1-) anions [45,48].)

Other points.

- The paper is too long and several characterization data could be supplied as supporting information (such as the elemental analysis data and the NMR chemical shift table). The Introduction chapter should be shortened, removing ‘trivial’ discussion points, such as the acid-base behavior of amino acids.

-        Necessary arrangements were made by evaluating the criticism of the referee. Table showing 11B-NMR shear values, the figure showing thermal analysis curves and figures showing mass analysis were added to the supplementary information section.

- Figure 9 (mass spectra) is too small and, therefore, not readable.

-        Mass analysis spectra were enlarged and added to supplementary information.

- There is no comparison with previously reported pentaborate salts cited in the introduction. It should be of interest to know the differences between the present compounds and pentaborate salts having metal cations as counter ions.

-        Necessary literature sources related to existing pentaborate structures are given.

- There almost no experimental data. For instance, which solvent used the authors to collect the 11B NMR spectra?  Did the authors collect NMR spectra on solid compounds?

-        On page 2, line 138 indicates that the solvent used in the 11B-NMR device information is D2O.

-        Furthermore, since there are no solid 11B-NMR device, 11B-NMR shift values of the samples in solid form could not be determined.

In summary, I cannot suggest the publication of this paper. The compounds should be carefully isolated and purified. The paper should be deeply revised, removing some not useful details and adding some discussion/comparison with similar compounds. Finally, the language used should be deeply revised and the paper should be corrected by a native English speaker.

-        Considering all the criticisms made by the referee, the proposed arrangements, additions and changes have been made.

Reviewer 3 Report

My comments are in attached file.

Author Response

1. All suggestions and corrections made by the referee were made on the text.

2. 6 references (3-6) and (11,12) recommended by the referee were annexed to the article.

3. Significantly distressed sentences have been renewed.

Round 2

Reviewer 1 Report

Obviously, the authors do not understand proper English! For example, both words "Cations  (NMCs) and Pentaborates are ''objects"  and it should be combined to form one "object". Therefore, the TITLE should read as "Amino Acid Derivatives as Novel Non-Metal Cations (NMCs) of Pentaborate Salts"! The English in the text has not been improved and it must be corrected before publication.  I, therefore, recommend a "major revision".

Author Response

The English version of the text has been reviewed. In particular, the title has been corrected.

Reviewer 2 Report

The paper has been revised by the authors following the suggestions of the reviewer and the overall quality has been improved.   However, it still contains some problems in English style/grammar, which should be improved by the authors or by the Editor.

Author Response

That's ok!

Reviewer 3 Report

After the revision, the paper is greatly improved and, in my opinion, it can be published in the present state.

Author Response

That's ok!